# Controlled and customizable baculovirus NOS3 gene delivery using PVA-based hydrogel systems

Sabrina Schaly[1], Paromita Islam[1], Jacqueline L. Boyajian[1], Rahul Thareja[1], Ahmed Abosalha[1,2], Karan Arora[1], Dominique Shum-Tim[3], Satya Prakash[1]*

1 Biomedical Technology and Cell Therapy Research Laboratory, Department of Biomedical Engineering, Faculty of Medicine and Health Sciences, McGill University, Montreal, Quebec, Canada, 2 Pharmaceutical Technology Department, Faculty of Pharmacy, Tanta University, Tanta, Egypt, 3 Division of Cardiac Surgery, Royal Victoria Hospital, McGill University Health Centre, McGill University, Faculty of Medicine, Montreal, Quebec, Canada

* satya.prakash@mcgill.ca

## Abstract

Nitric oxide synthase 3 (NOS3) eluting polyvinyl alcohol-based hydrogels have a large potential in medical applications and device coatings. NOS3 promotes nitric oxide and nitrate production and can effectively be delivered using insect cell viruses, termed baculoviruses. Nitric oxide is known for regulating cell proliferation, promoting blood vessel vasodilation, and inhibiting bacterial growth. The polyvinyl alcohol (PVA)-based hydrogels investigated here sustained baculovirus elution from five to 25 days, depending on the hydrogel composition. The quantity of viable baculovirus loaded significantly declined with each freeze-thaw from one to four (15.3 ± 2.9% vs. 0.9 ± 0.5%, respectively). The addition of gelatin to the hydrogels protected baculovirus viability during the freeze-thaw cycles, resulting in a loading capacity of 94.6 ± 1.2% with sustained elution over 23 days. Adding chitosan, PEG-8000, and gelatin to the hydrogels altered the properties of the hydrogel, including swelling, blood coagulation, and antimicrobial effects, beneficial for different therapeutic applications. Passive absorption of the baculovirus into PVA hydrogels exhibited the highest baculovirus loading (96.4 ± 0.6%) with elution over 25 days. The baculovirus-eluting hydrogels were hemocompatible and non-cytotoxic, with no cell proliferation or viability reduction after incubation. This PVA delivery system provides a method for high loading and sustained release of baculoviruses, sustaining nitric oxide gene delivery. This proof of concept has clinical applications as a medical device or stent coating by delivering therapeutic genes, improving blood compatibility, preventing thrombosis, and preventing infection.

## Introduction

Polyvinyl alcohol (PVA) is one of the most versatile synthetic polymers used in medical applications. PVA has several applications, including tissue engineering, dental implants, drug delivery systems, wound dressings, contact lenses, cosmetics, and more [1]. PVA's popularity may be attributed to its inherent biosafety and hemocompatibility [2]. Even when administered orally, PVA demonstrates low toxicity (a lethal dose of 15–20 g/kg), is poorly absorbed in

**Data Availability Statement:** All relevant data are within the paper and its Supporting information files.

**Funding:** This work was supported by a grant from the Canadian Institute of Health Research (CIHR, grant 252743) to Dominique Shum-Tim and Satya Prakash. S.S. is fully funded by the Canadian Graduate Scholarship-Doctoral from the Natural Sciences and Engineering Research Council (NSERC, 569661-2022). P.I. is funded by the Islamic Development Bank Scholarship (2020-245622). A.A. is fully funded by a scholarship from the Ministry of Higher Education of the Arab Republic of Egypt. R.T. and K.A. are funded by the Canadian Graduate Scholarship-Master's from NSERC. The funders had no role in study design, data collection and analysis, decision to publish, or preparation of the manuscript.

**Competing interests:** The authors have declared that no competing interests exist.

the gastrointestinal system, and is not mutagenic [3]. Another benefit of PVA is the tunability of its mechanical properties. Specifically, the tensile strength, toughness, elongation, and elastic modulus can vary significantly depending on the ions used during the aggregation [4]. The properties of the PVA can be further customized by adding other polymers or particles. Previously, chitosan, cellulose, Gellan Gum, protein, starch, nanotubes, and nanoparticles, and more have been used to alters PVA's properties [5]. Moreover, the solution temperature and synthesis method (ex., cross-linker) alter the polymer's strength and elasticity. These modifiable properties are beneficial when customizing hydrogels for different medical applications.

The addition of different polymers to PVA also provides new biological properties. For example, gelatin is a natural polymer that promotes cell attachment in tissue engineering applications and cell spheroids [6]. Gelatin is also non-toxic, non-immunogenic, low-cost, and widely available [7]. Gelatin has been investigated and used in drug delivery carriers, bioinks, tissue engineering, and wound dressings. Alternatively, chitosan has anti-inflammatory, antimicrobial, and antioxidant properties [8]. Also, chitosan can bind negatively charged particles, including baculoviruses to improve transduction efficiency. Adding chitosan has been shown to increase the tensile strength and elastic modulus while decreasing the elongation of PVA-chitosan films [5]. Finally, polyethylene glycol (PEG) was selected for its biocompatibility and controlled drug release properties [9].

Hydrogels are helpful delivery platforms for drugs, proteins, genes, or growth factors to treat several diseases. PVA-based hydrogels have been studied for treating colon cancer, sustained ibuprofen delivery, and medical device coatings [10–12]. Specifically, nitric oxide-releasing coatings are beneficial for their enhanced biocompatibility, inhibition of thrombosis, and antibacterial properties. Moreover, nitric oxide inhibits platelet adhesion and reduces smooth muscle cell proliferation, which are beneficial hemocompatibility properties [12]. Nitric oxide also has antimicrobial effects and prevents biofilm creation [13, 14]. These properties make PVA hydrogels promising for medical device coatings, specifically stent coatings.

Stenting is a common treatment for coronary artery disease. However, damage to the arterial cells, mainly endothelial and smooth muscle cells (SMCs), leads to complications such as in-stent restenosis (ISR) and thrombosis. Different stent coatings have been developed, including drug-eluting stents to prevent inflammation and ISR. Alternatively, nitric oxide delivery regulates blood coagulation, prevents platelet adhesion, prevents thrombosis, prevents SMC proliferation, and promotes endothelial health and migration for re-endothelialization [15]. NOS3 also plays a key role in vascular homeostasis and protects the intima from platelet aggregation and leukocyte adhesion [16]. NOS3 expression is crucial in the initial two weeks after stent implantation to regulate vascular function and promote an anti-inflammatory environment [15]. Controlled NOS3 delivery could thus prevent platelet aggregation and SMC hyperproliferation, mitigating complications such as thrombosis and in-stent restenosis, respectively.

There are challenges associated with eluting genes from hydrogels. Some limitations include burst release, growth factor degradation, and low loading efficiency. One limitation associated with PVA hydrogels, is their hydrophilicity, making hydrophobic drug loading challenging and leading to an initial burst release [17]. However, this limitation can be overcome by optimizing the synthesis method, adding additional polymers, or an additional coating [17, 18]. One method to extend and control gene elution is through a viral delivery system. Specifically, baculoviruses are insect cell viruses that allow transient gene delivery in humans [19]. Baculoviruses are also non-toxic, have low immunogenicity, and provide transient yet high gene expression [20–25]. Compared to other viral vectors such as adenoviruses, retroviruses, and lentiviruses, baculoviruses have lower immunogenicity, lower cost, higher biosafety, larger cloning capacity (300+ kb), and are easier to scale up [24]. Baculoviruses also

demonstrate higher gene expression than direct gene delivery due to the protective virus capsid [25]. However, baculovirus stability has been reported to decline quickly at temperatures matching or exceeding room temperature [26]. To mitigate this, the PVA hydrogel delivery system was developed to prevent baculovirus decline and provide controlled release. In addition, baculovirus encapsulation within hydrogels can prevent immune inactivation, effectively shielding the virus from the immune system and promoting its stability. The polymer gene delivery platform also promotes site-specificity by releasing virus directly at the contact site [27]. The site specificity allows for a lower dosage and improves the therapeutic range.

Here, we develop and investigate a PVA-based baculovirus eluting delivery system. We will investigate how PVA-based hydrogels can overcome limitations associated with baculovirus gene delivery. Different hydrogel synthesis methods are explored to determine their impact on virus stability and loading efficiency. Moreover, polymers are added to the PVA hydrogels to modify PVA's properties. Specifically, we investigate how the elution duration, hydrogel properties, and gene dosage can be controlled and customized. Controlled baculovirus elution studies have been limited [28]. To our knowledge, this is the first study optimizing PVA-based hydrogels containing baculovirus gene delivery vectors. This is used as a proof of concept that PVA-baculovirus hydrogels can be tunable to treat a wide variety of diseases and are a promising option for medical device coating.

## Materials and methods

### Hydrogel synthesis

A 20% (m/v) solution of PVA (MW 60,000 from Sigma Aldrich) was made by dissolving PVA in distilled water at 100 ˚C and 500 rpm. Meanwhile, a 5% (m/v) gelatin solution was made in PBS. A 1 mg/mL chitosan solution was made in PBS with 0.5% (v/v) glacial acetic acid. The three solutions were autoclaved at 120 ˚C for 15 minutes and then cooled to 37 ˚C. An equal amount of PVA was poured into each well of a 48-well plate under aseptic conditions. Next, 100 μL of the 1% (m/v) gelatin, 1 mg/mL chitosan, or 0.1% (v/v) PEG 8000 (Thermo Fisher) was added to the designated wells to create a PVA-gelatin, PVA-chitosan, or PVA-PEG hydrogels, respectively. The PVA-based hydrogels were stirred with a pipette tip to mix. Next, the baculovirus stock solution was added to each well and stirred a final time. PBS was added to hydrogels to serve as a mock. The hydrogels were frozen overnight and thawed for two hours to constitute one freeze-thaw cycle. The freeze-thaw cycle was repeated one to four times depending on the hydrogel.

### Attenuated Total Reflectance- Fourier Transform Infrared (ATR-FTIR) spectroscopy

The spectra of the different hydrogels were obtained using an FTIR Spectrum II (Perkin Elmer, Waltham, Massachusetts, USA) with ATR. The data were normalized to the background spectrum without a hydrogel. Next, the hydrogels were clamped down and a spectral range of 400–4000 cm$^{-1}$ was recorded with four scans and a resolution of 1 cm$^{-1}$.

### Hydrogel swelling and degradation

The hydrogels were dried overnight at 37 ˚C. The next day the hydrogels were weighed, and 1 mL of Hanks Balanced Salt Solution (HBSS) was added to each hydrogel. The hydrogel was removed from the HBSS at the designated time points and weighed. The swelling ratio was

calculated using the following equation:

$$\text{Swelling ratio} = \text{Weight}_{swollen}/\text{Weight}_{dry}$$

### Disc diffusion antimicrobial studies

The antimicrobial properties of the PVA-based hydrogels were tested using an adapted disc diffusion method [29]. *E. coli* and *C. Albicans* were grown in Luria-Bertani (LB) and yeast extract peptone dextrose (YPD) media, respectively. The number of colony-forming units (CFU) was calculated by plating dilutions of the bacteria and incubating overnight. Based on the calculated CFU, 100 μL of $10^8$ CFU/mL was spread onto fresh LB or YPD agar plates. The PVA-based hydrogels were plated on top of the bacteria and incubated 37˚C for 24 hours. A disc covered with penicillin (5 μL) was used as a negative control. After 24 hours, the area of growth inhibition was observed and measured.

### Scanning electron microscopy (SEM)

The PVA-hydrogels were dried overnight at 37˚C before being imaged. A FEI 450 Quanta SEM was used at the Facility for Electron Microscopy Research at McGill University. A high vacuum and a voltage of 10 kV were applied.

### Cell culture

Human umbilical vein endothelial cells (HUVECs) and smooth muscle cells (SMCs) were maintained in T-25 flasks in a 37 ˚C, 5% $CO_2$ incubator. The HUVECs were purchased from Sigma Aldrich and the SMCs from Coriell Institute for Medical Research (Camden, NJ, USA). Medium 199 was used with 0.02 mg/mL endothelial growth supplement, 10% fetal bovine serum was used to culture the SMCs, and complete endothelial growth medium (Sigma Aldrich, Burlington, MA, USA) was used to culture the HUVECs. Cells were used within five passages upon receiving.

### Insect cell culture

Sf9 insect cells (Sigma Aldrich, Burlington, MA, USA) were maintained at 27 ˚C in BacVector medium (Sigma Aldrich) in T-75 flasks or 250 mL shake flasks (Erlenmeyer, CA). The Sf9 cells were subcultured 2–3 times weekly and shaken at 130 rpm.

### Gene selection

The NOS3 gene was purchased from GenScript (Piscataway, NJ, USA). and cloned into a pOET6 plasmid (MJS Biolynx Inc., Brockville, ON, Canada) containing the cytomegalovirus (CMV) promoter. GenScript also provided a pBackPak9 plasmid expressing EGFP and RFP under a polyhedron and CMV promoter, respectively. The concentration and purity of the plasmid were confirmed using a NanoDrop (Thermo Fisher, Waltham, MA, USA) and were used for all future virus production steps.

### Baculovirus production

The supplier's protocol was followed, as previously described, for baculovirus production [30]. In an exponential growth phase, $5 \times 10^5$ S9 cells were seeded into a 12-well plate one hour before virus transfection. Next, 200 ng of the pBakPak9 plasmid was added to 100 ng of flash-BAC DNA (MJS Biolynx Inc., Brockville, ON, Canada), 0.48 μL of TransIT Insect Reagent,

and 100 μL PBS and incubated at room temperature for 15 minutes. The transfection mixture was then added to the Sf9 cells and incubated overnight at 27 ˚C. The next day, 0.5 mL of Bac-Vector media was added. Five days after transfection, the culture medium was harvested, centrifuged at 1000 x g for 10 minutes, and stored at 4 ˚C. This generated the $P_o$ virus stock and was added to 100 mL of Sf9 cells at a concentration of $2x10^6$ cells/mL. The infected Sf9 cells were agitated at 130 rpm for four days before harvesting the culture medium as described above. This generated $P_1$ baculovirus stock expressing an enhanced green fluorescent protein (EGFP) or NOS3.

## Baculovirus titration

The EGFP-baculovirus was titrated using a fluorescent tittering assay. First, $5 \times 10^5$ Sf9 cells/ well were seeded into a 12-well plate and incubated for one hour. During the incubation, the virus was diluted using insect cell medium (1:10, 1:25, 1:50, 1:75, 1:100). The cell media was then aspirated off the Sf9 cells and 100 μL of the virus dilution was added to each well. The plate was incubated for one hour and then an additional 0.5 mL of insect cell media was added to each well and incubated at 27 ˚C. Forty-eight hours after, the number of fluorescent cells was counted in each well with less than 40% of fluorescent cells. The following equation was used to calculate the titration:

$$\text{Transduction Units (TU)/mL} =$$

$$(\text{number of cells transduced x percent fluorescent x dilution factor})/$$

$$(\text{transduction volume in mL})$$

The NOS3-baculovirus was titrated using the FastPlax Titer kit from Sigma Aldrich as specified by the supplier.

## Baculovirus elution from the hydrogels

The hydrogels were incubated in 1 mL of Hanks Balanced Salt Solution (HBSS). All the HBSS was removed and replaced with fresh HBSS at the designated times. The amount of baculovirus in the HBSS was quantified using the Fluorescent tittering assay described above. The HBSS containing the baculovirus was also used to transduce mammalian cells.

## Baculovirus transduction

The hydrogel suspension media (virus in HBSS) was removed and used to transduce mammalian cells (SMCs and HUVECs) in 96-well plates. After three hours of incubation, the virus inoculum was removed and replaced with fresh cell media. The cells were incubated for 48 hours to allow for viral gene expression. Fluorescent images of the cells were taken at different time points and stained with DAPI at 48 hpi. Images were acquired using the Columbus Image Data Storage and Analysis System (PerkinElmer, USA) and the Operetta® CLS automated microscope. Image acquisition (Fluorescence) and analysis (Columbus and Harmony) was performed with the McGill University Imaging and Molecular Biology Platform (IMBP) equipment or services.

## Reverse transcription-quantitative polymerase chain reaction (RT-qPCR)

To perform RT-qPCR, cells were transduced as described above. Twenty-four hours after transduction, the cell RNA was extracted using an RNA extraction kit (Bio Basic) and stored at -80 ˚C until use. Luna Universal One-Step RT-qPCR kit (New England Biolabs, Ipswich, MA, USA) was used to reverse transcribe the stored RNA. Primers for NOS3 and B-actin reference

genes were purchased from Bio-Rad (Hercules, CA, USA) and mixed with the RNA and primers at the supplier-defined concentrations. The reaction was run on an Illumina Eco Real-Time PCR machine (San Diego, CA, USA) as directed by the supplier (NEB).

## MTT proliferation assay

An MTT proliferation assay (Sigma Aldrich, St. Louis, MI, USA) was used. As above, different baculovirus MOIs were added to HUVEC and SMCs and incubated for three hours. After three hours of incubation, the baculovirus supernatant was removed and replaced with fresh cell media. For the PVA-based hydrogels, cells were seeded into 24 well plates containing the hydrogels and incubated for 48 hours.

For the MTT assay, 0.01 mL of AB Solution (MTT) was added to each well 48 hpi. The cells were incubated for two hours at 37 ˚C to allow MTT cleavage. After two hours, 0.1 mL of isopropanol with 0.04 N HCl was added to each well. The isopropanol solution was mixed thoroughly via pipetting. The plate was then read using an EnSpire Multimode plate reader (Perkin Elmer, USA) with a wavelength of 570 nm.

## Live dead assay

A Live Dead Assay was employed to estimate the hydrogels' cytotoxicity using Calcein AM and propidium iodide (Thermo Fisher, Waltham, MA, USA). A 96-well plate was seeded with cells, as described above, and transduced with different MOIs of the baculovirus. DMSO and regular cell media were used as the controls.

For the hydrogels, $2 \times 10^4$ HUVECs/well were seeded into a 48-well plate and incubated overnight. The next day, the PVA hydrogels were added to each well and incubated for 24 to 48 hours. Media alone or DMSO was used as a control.

After 48 hours, 5 μL of 1 mM Calcein AM and 5 μL of 2.5 mg/mL propidium iodide were added to 10 mL of cell media. This solution was then added to the cells in the 48-well or 96-well plates and incubated at 37˚C for 30 minutes. After 30 minutes, the cells were imaged using the Operetta High-Content Analysis system and the percent of live cells was calculated using Columbus (Perkin Elmer).

## Griess assay

A griess assay kit was used to measure nitrate production following the supplier's instruction (Thermo Fisher, Walktham, MA, USA). A standard curve was used to calculate nitrate concentration in HUVECs after NOS3-baculovirus transduction.

## Antibacterial assay for NOS3 and nitrate

The antibacterial properties of NOS3 and nitrate were investigated using broth dilutions to determine the minimal inhibitory concentration. *E. coli* from the hydrogel antimicrobial studies was used. First, the bacteria were diluted to $10^6$ CFU/mL and 100 μL was added to each well of a 96-well plate. Next, 100 μL of the cell media supernatant containing NOS3 or sodium nitrate (10, 5, and 2.5 μM) was added to the wells. Cell media without NOS3 and HBSS was added as the control. The plate was read on the HTS7000 microplate reader every two hours for 8 hours to create a bacterial growth curve.

## Hemocompatibility

Two blood samples were purchased from Innovative Research (Novi, MI, USA). A standard hemolysis assay was performed by immersing all samples into 5 mL of PBS in a 15 mL

centrifuge tube. Next, 4 mL of citrated blood was added to 5 mL PBS. To each sample, 0.1 mL of diluted blood was added. The samples were incubated at 37°C for one hour and then centrifuged at 1,000 rpm for 10 min. All supernatants were placed into a 96-well plate and read at 545 nm. The negative and positive control was PBS and deionized water, respectively. The following equation was used to determine the percent hemolysis.

$$\text{Hemolysis}(\%) = \left(\left(\text{OD}_{\text{sample}} - \text{OD}_{\text{negative control}}\right)\Big/\left(\text{OD}_{\text{positive}} - \text{OD}_{\text{negative}}\right)\right) \times 100\%$$

Blood coagulation on hydrogels was evaluated using an adapted method by Sabino et al. (2020) [31]. Briefly, the hydrogels were placed into a 24-well plate and 7 μL of whole blood was pipetted onto the surface. After the designated time point, deionized water was added to each well for 5 minutes. The hydrogel supernatant was added to a 96-well plate and read at 540 nm. For controls, 7 μL of whole blood was added to distilled water and a 'mock' polystyrene surface was also used to compare the blood coagulation properties of the hydrogels.

### Statistical analysis

Unless otherwise stated, all results are reported as a mean ± standard deviation (SD) in triplicates. GraphPad Prism (version 9.0.0 for Windows, GraphPad Software, San Diego, California USA, www.graphpad.com) was used for all one-way ANOVA data analysis and graph creation. Ethics approval was not required for the study.

## Results and discussion

### Morphology, swelling, antibacterial, and hemocompatibility of the PVA-based hydrogels

First, freeze-thawed PVA hydrogels were investigated for sustained baculovirus delivery. Up to four freeze-thaw cycles were used. For one, two, three, and four freeze-thaw cycles, the hydrogels are named PVA1, PVA2, PVA3, PVA4, respectively. The number of freeze-thaw cycles was optimized to maximize baculovirus entrapment. The functionality of baculoviruses diminished with every freeze-thaw cycle. However, to maintain mechanical strength, two freeze-thaw cycles were used. Consequently, unless otherwise stated, two freeze-thaw cycles were employed for all subsequent hydrogels. Next, different polymers were added to the PVA solution before casting, which altered the baculovirus entrapment and release. Chitosan, PEG-8000, or gelatin were added to the PVA solution, termed PVA-chitosan, PVA-PEG, and PVA-gelatin, respectively. After this, the baculovirus was added and the solution mixed. After two freeze-thaw cycles, the hydrogels were removed and incubated in HBSS to test the virus elution over time.

The properties of the PVA-based hydrogels were evaluated via ATR-FTIR, swelling ratios, antimicrobial properties, and surface morphology using scanning electron microscopy (SEM). The addition of each polymer was confirmed with ATR-FTIR, with new peaks indicating the presence of gelatin, chitosan, or PEG-8000 (Fig 1). The PVA base had the characteristic FTIR peaks for the hydroxyl group (3200–3600 cm$^{-1}$), C-H stretching (2850–3000 cm$^{-1}$), the carbonyl group (1745 cm$^{-1}$), and CH$_2$ wagging (1400 cm$^{-1}$). Gelatin also had a characteristic peak due to N-H stretching (3200–3600 cm$^{-1}$). The amide bond within the gelatin also added an additional peak at 1540–1650 cm$^{-1}$. PEG-8000 added peaks for the hydroxyl groups and C-H stretching. The ether bond within PEG also added a peak around 1100–1200 cm$^{-1}$.

The hydrogels initially swelled up to three times their weight after incubating in HBSS for one hour. After two hours, most hydrogels had reached their peak swelling capacity (Fig 1). The PVA-based hydrogels had different swelling ratios depending on the polymer addition.

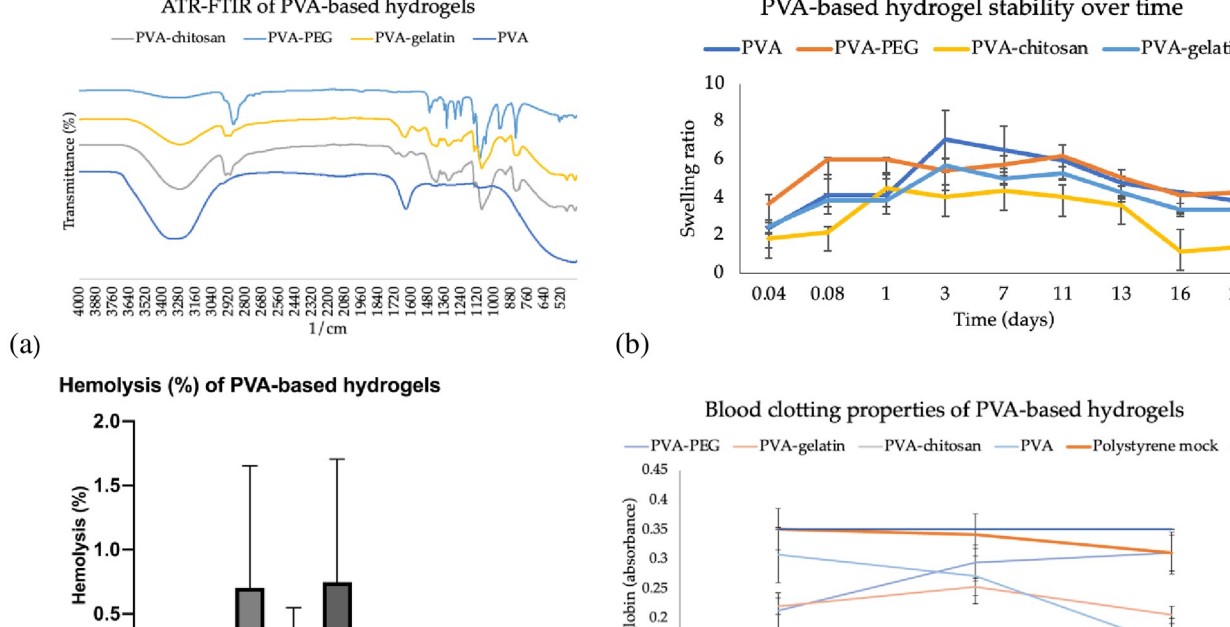

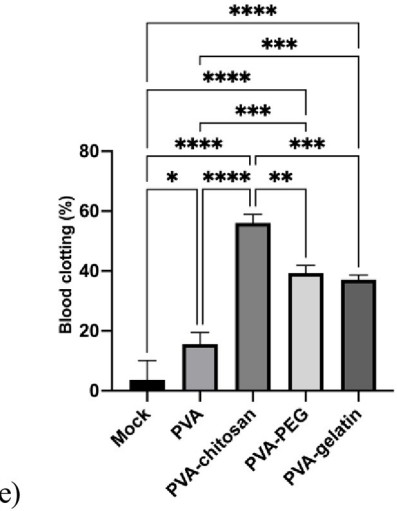

**Fig 1. Properties of the PVA-based hydrogels.** (a) ATR-FTIR of PVA-based hydrogels. (b) PVA-based hydrogel swelling over time. (c) Hemolysis of red blood cells after incubation with PVA-based hydrogels. (d) Blood coagulation on the surface of the PVA-based hydrogels after incubating over time. (e) Blood coagulation on the surface of the PVA-based hydrogels after 45 minutes of incubation.

The maximum swelling ratio varied from 4.0 to 7.1 (Table 1). The swelling increased over time, peaking at day three. After day three, the swelling ratio slowly decreased until the original weight was achieved from day one indicating suitable stability over 21 days. Passive baculovirus diffusion onto the PVA hydrogel was the most efficient and thus the stability was

**Table 1. PVA-based hydrogel swelling ratio and antimicrobial properties.**

| Hydrogel type | Maximum swelling ratio (average ± SD) | Antimicrobial inhibition (mm ± SD) |
|:---:|:---|:---|
| PVA | 7.1 ± 1.5 | 2.0 ± 0.1 |
| PVA-PEG | 6.0 ± 0.2 | 2.0 ± 0.1 |
| PVA-chitosan | 5.7 ± 3.4 | 3.0 ± 0.1 |
| PVA-gelatin | 4.0 ± 0.6 | 2.0 ± 0.1 |

evaluated over two months. Overall, no significant change in weighted stability was observed over the 56 days tested at accelerated conditions (37 ˚C and 150 rpm). The antimicrobial properties of the hydrogels were also investigated (Table 1). All hydrogels blocked bacterial growth directly beneath (2.0 ± 0.1 mm), but only PVA-chitosan resulted in a one-millimeter growth inhibition zone of *E. coli* (3.0 ± 0.1 mm) but not *C. Albicans*. This was compared to a growth inhibition zone of 3.0 ± 0.5 mm using 500 units of penicillin-streptomycin control solution.

The PVA-based hydrogels also demonstrated hemocompatibility. The blood compatibility was evaluated using a standard hemolysis and blood coagulation assay. After the one-hour incubation, some degree of hemolysis was observed after incubating the blood samples with the hydrogels (Fig 1c). However, all the hydrogels were considered non-hemolytic with a hemolysis below 2.0%. The PVA-chitosan and PVA-PEG hydrogels produced the largest variation in hemolysis. Comparatively, the PVA and PVA-gelatin hydrogels demonstrated the least hemolytic potential (below 0.5%).

Blood coagulation properties were also investigated. First, blood was also added directly to the surface of the hydrogels and incubated for 15 to 45 minutes. Each hydrogel composition impacted blood clotting differently (Fig 1d and 1e). Compared to the polystyrene mock control, significantly more blood clotted to the PVA hydrogel (p<0.05). Blood coagulation was also significantly higher (p<0.00005) for the PVA-gelatin, PVA-PEG, and PVA-chitosan hydrogels compared to the mock control. Overall, the PVA hydrogel showed the least blood coagulation (15.5 ± 3.8%) and PVA-chitosan showed the most (56.0 ± 2.85%), as indicated by the limited free hemoglobin after washing the hydrogels with deionized water. All PVA-based hydrogels maintained a similar smooth appearance under SEM despite adding other polymers (Fig 2). These properties could be used to customize the hydrogels for specific applications. For example, the PVA-chitosan hydrogel prevented E. Coli growth (antibacterial) and promoted blood coagulation. The PVA-chitosan hydrogel was also capable of swelling four-fold to maintain a moist environment, indicating its suitability for wound dressing applications [32, 33]. Comparatively, the PVA hydrogel creates a smooth surface with minimal blood coagulation, indicating a potential use for vascular grafts or other applications requiring an anti-thrombotic polymer [34]. Finally, the PVA and PVA-gelatin hydrogels demonstrated the best baculovirus retention and sustained delivery making them useful for long term applications such as medical device or stent coatings [35].

## Baculovirus production and elution from PVA-based hydrogels

We created two different baculoviruses. One baculovirus expressed EGFP in insect cells and RFP in mammalian cells. The second baculovirus expressed nitric oxide synthase three (NOS3) in mammalian cells. The baculovirus was amplified in 100 mL of Sf9 cells in BacVector media. The virus supernatant was collected and titrated using a fluorescent tittering assay or FastPlax titration kit with dilutions for an accurate estimate. The Sf9 cells were imaged 48 hours after infection, at the first sign of EGFP expression. The final titration was calculated to

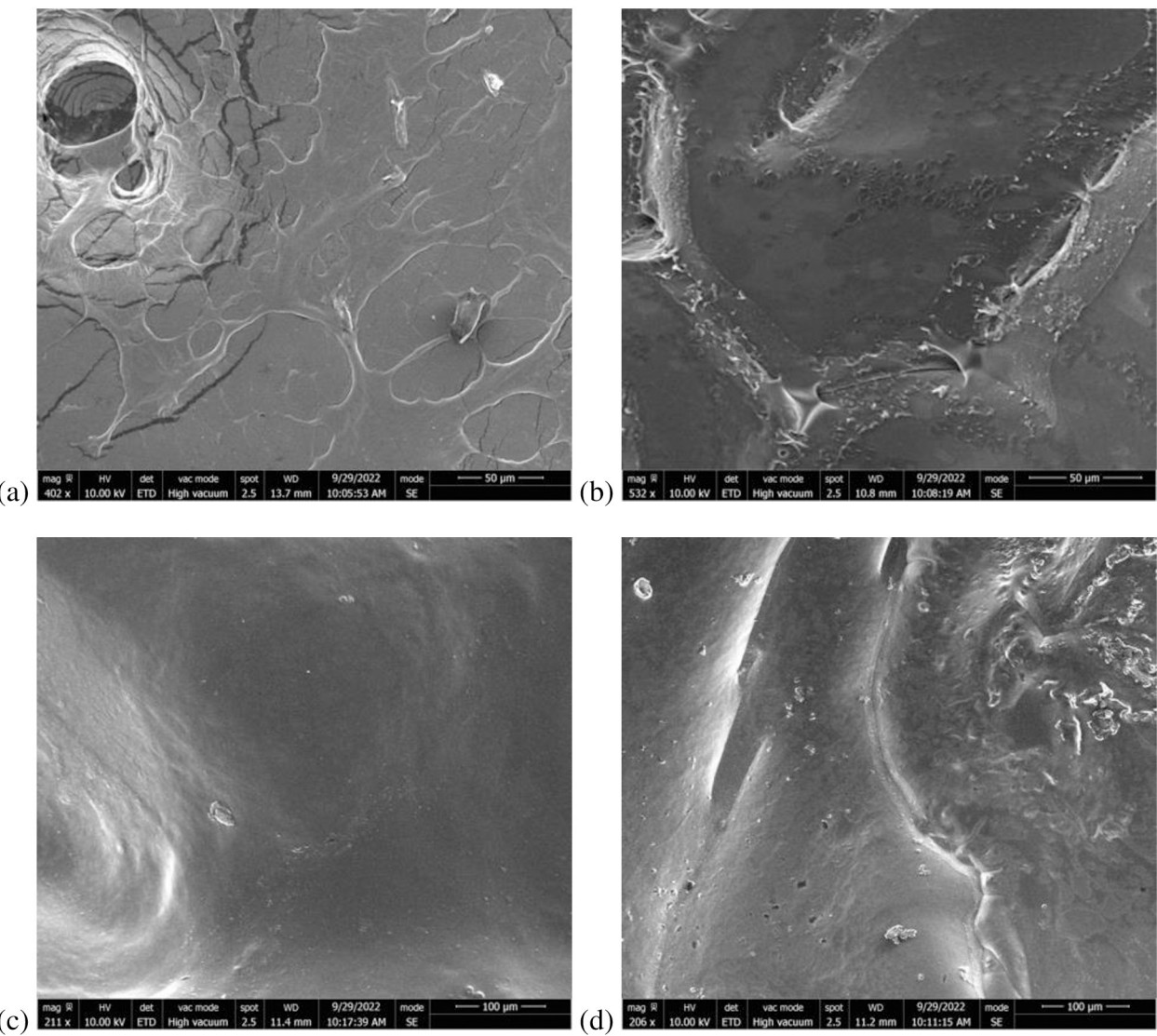

**Fig 2. SEM of different PVA-based hydrogel compositions.** (a) PVA, (b) PVA-PEG, (c) PVA-chitosan, and (d) PVA-gelatin.

be 1.4 x $10^8$ TU/mL for the EGFP-baculovirus and 1.0 x $10^8$ PFU/mL for the NOS3-baculovirus.

Before the freeze-thaw cycles, the baculovirus stock was added directly to the PVA-based hydrogels. After the designated freeze-thaw cycles, the hydrogels were removed from the casting well plate and incubated in HBSS. At the designated time points, the HBSS was removed and added to Sf9 cells to determine the baculovirus elution over time. The baculovirus viability significantly decreased with each freeze-thaw cycle (Fig 3). The hydrogel loading efficiency varied for each freeze-thaw cycle, ranging from 15.3 ± 2.9% to 0.9 ± 0.5% (Table 2).

The addition of different polymers to the PVA hydrogel improved baculovirus entrapment and elution. Specifically, adding PEG to the PVA hydrogel prolonged the baculovirus elution compared to the PVA2 hydrogel. The majority of the virus was eluted between days three to five for the PVA-PEG hydrogel. Similarly, adding chitosan to the PVA seemed to reduce the

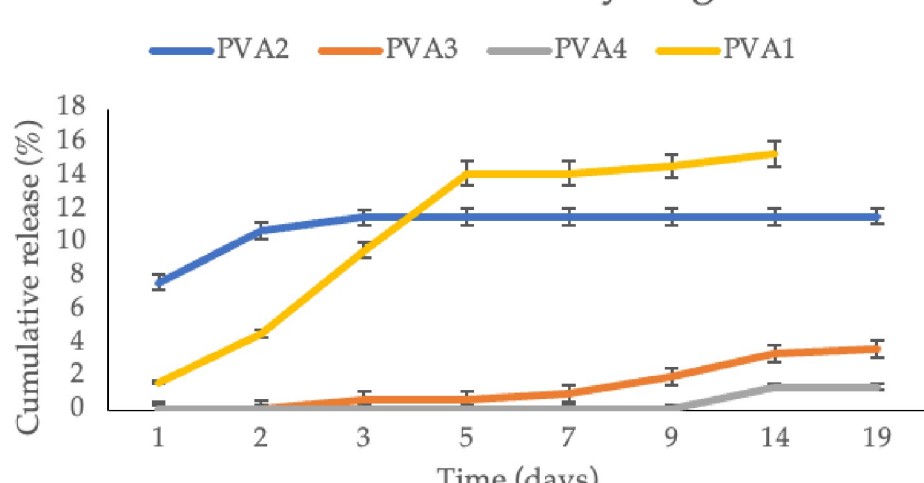

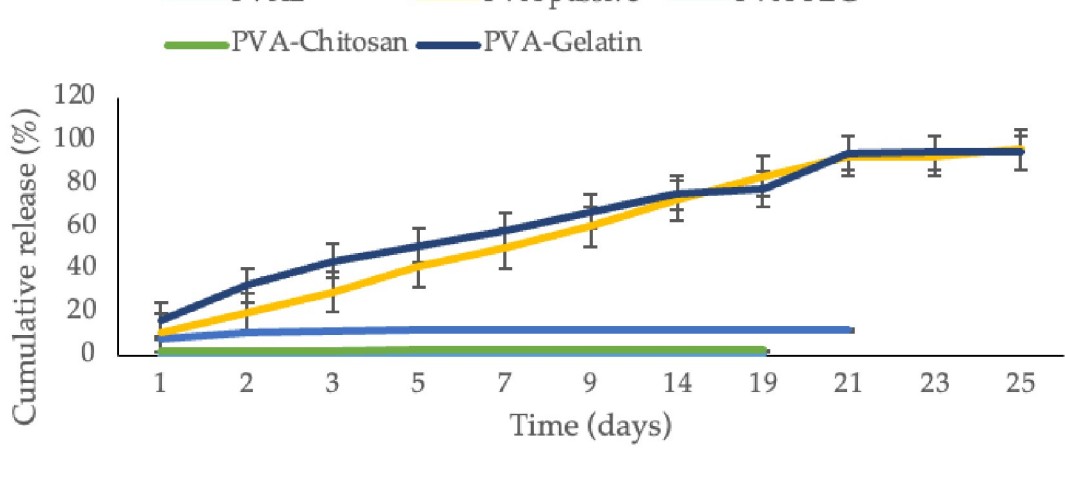

**Fig 3. Baculovirus elution from PVA-based hydrogels over time.** (a) BV elution from PVA hydrogels based on the number of freeze-thaw cycles. (b) BV elution from different PVA-based hydrogel compositions.

amount of baculovirus entrapped and released over time and may be attributed to chitosan's binding to negatively charged virus particles. Comparatively, adding gelatin significantly improved baculovirus retention and controlled release over time compared to all other freeze-thawed PVA-based hydrogels.

Once all the baculovirus was eluted, the total amount was used to determine the hydrogel loading efficiency. The loading efficiency varied from 0.4 ± 0.7% to 94.6 ± 1.2% depending on the hydrogel composition, listed in Table 2. For most of the hydrogels, the baculovirus was eluted in the first three days, such as for PVA2 and PVA-chitosan. The freeze-thawed PVA-gelatin hydrogel and the PVA hydrogel loaded passively with baculovirus demonstrated the most baculovirus elution (Fig 3). Specifically, the passive absorption of baculovirus into PVA

**Table 2. PVA-based hydrogel loading efficiencies.**

| Hydrogel | Loading efficiency (% ± SD) |
|---|---|
| PVA1 | 15.3 ± 2.9 |
| PVA2 | 11.8 ± 1.0 |
| PVA3 | 3.7 ± 2.1 |
| PVA4 | 0.9 ± 0.5 |
| PVA passive absorption | 96.4 ± 0.6 |
| PVA-PEG | 0.4 ± 0.7 |
| PVA-chitosan | 2.3 ± 1.7 |
| PVA-gelatin | 94.6 ± 1.2 |

hydrogels resulted in a loading efficiency of 96.4 ± 0.6%. Similarly, the PVA-gelatin hydrogels had a loading efficiency of 94.6 ± 1.2%. The elution from these hydrogels also lasted over 23 days.

The addition of polymers (chitosan, gelatin, and PEG) to PVA hydrogels controlled baculovirus elution and loading. Baculovirus elution was sustained with all hydrogels. The elution time varied from 5 to 25 days. All hydrogels exhibited a slow first-order release of the baculovirus that eventually plateaued. With each freeze-thaw cycle, the amount of viable baculovirus within each hydrogel diminished significantly, consistent with previous results [36]. The amount of baculovirus encapsulated varied between freeze-thaw cycles as seen in Table 2. The addition of gelatin preserved the baculovirus viability during the freeze-thaw cycles compared to the other hydrogels. This supports previous studies where a gelatin coating maintains transfection efficiency and DNA stability [37, 38]. The baculovirus viability in PVA-gelatin hydrogels was comparable to the hydrogels loaded passively with baculovirus, not exposed to any freeze-thaw cycles. Passive loading of baculoviruses into PVA hydrogels exhibited the highest loading efficiency with sustained release. When $2 \times 10^7$ TU of baculovirus were added, the loading efficiency was 96.4 ± 0.6%. When $4 \times 10^7$ was loaded onto the PVA hydrogel, the loading efficiency dropped to 88.3 ± 0.6%. However, given the higher initial concentration, more baculovirus was loaded onto the second hydrogel overall. The different baculovirus doses and release times observed here can be customized to fit the medical application. For example, wound healing may require a sustained gene delivery over a couple of days, whereas PVA-coated stents require sustained delivery over approximately two weeks.

The optimal storage conditions for the baculovirus-eluting hydrogels include storing the virus in the dark at 4 °C for a couple months or at -80 °C for long term storage to preserve the virus titer and viability [39]. Similarly, the hydrogels can be stored at -20 °C or dried at 4 °C with no loss in weight after a month. Storing the baculovirus loaded hydrogel is less ideal as the baculovirus elutes over time as demonstrated in Fig 3.

## NOS3 delivery from eluted baculoviruses

The different PVA-based hydrogels were incubated in HBSS, and the supernatant was removed and replaced with fresh HBSS at the designated time points. The collected supernatant was added to Sf9 insect cells and used to transduce HUVECs to determine baculovirus release and activity over time. The baculovirus demonstrated efficient gene expression in both SMCs and HUVECs. As the amount of virus added (MOI) increased, the number of fluorescent cells or NOS3 expression also increased. The EGFP expression is shown below (Fig 4a). Similarly, the NOS3-baculovirus increased NOS3 expression increased dose-dependently. The highest dose, an MOI of 1000, increased NOS3 levels 17-fold compared to the mock

**GFP-baculovirus release after one day**

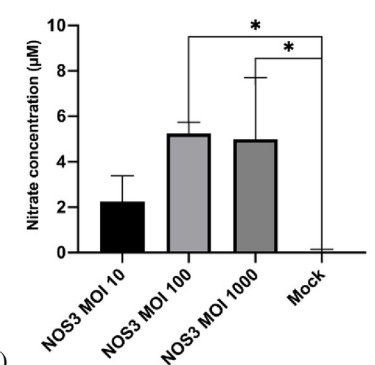

**NOS3 expression from eluted baculovirus**

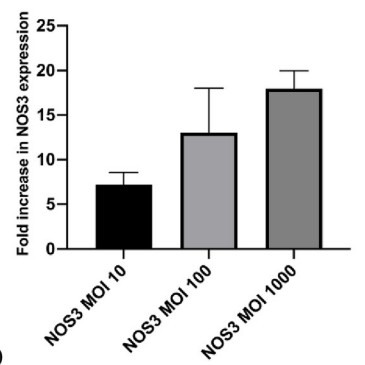

(a)

(b)

**Griess assay after baculovirus transduction**

**Antibacterial activity of NOS3 and nitrate**

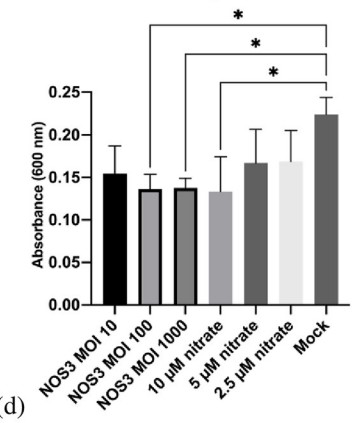

(c)

(d)

**HUVEC proliferation 48 hours after BV transduction**

**SMC proliferation 48 hours after BV transduction**

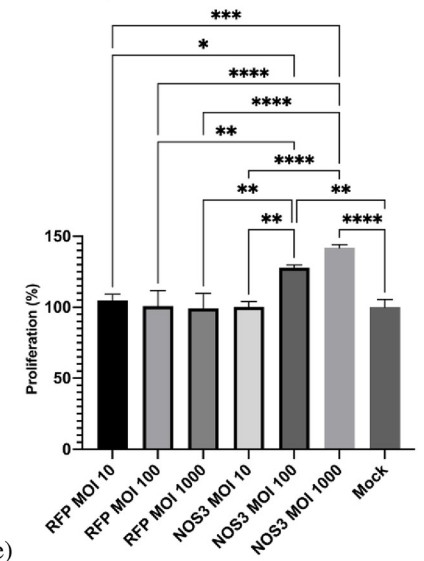

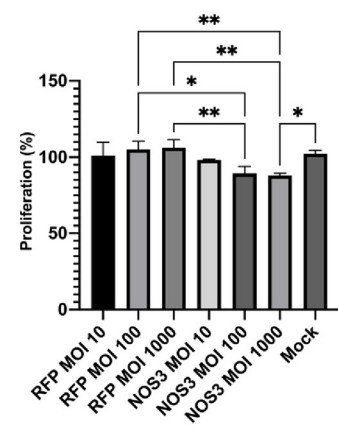

(e)

(f)

**Fig 4. Gene expression after baculovirus transduction.** (a) EGFP-baculovirus expression in Sf9 cells eluted from hydrogels after one day. The EGFP expression from a PVA-gelatin hydrogel is presented as a fluorescent image of DAPI-stained Sf9 cells (the scale bar represents 0.5 mm). (b) NOS3 expression based on baculovirus MOI. (c) Nitrate concentration after BV transduction. (d) Bacterial growth inhibition in the presence of NOS3 and nitrate. (e) HUVEC proliferation after BV transduction. (f) SMC proliferation after BV transduction.

transduced cells (Fig 4b). NOS3 promotes formation of nitric oxide and nitrate [40]. The media from the transduced cells was also used to measure nitrate formation (Fig 4c). Nitrate concentration was increased up to 8-fold compared to mock transduced cells. Nitric oxide and nitrate are effective antimicrobial agents against gram positive and gram negative bacteria preventing biofilm formation [41, 42]. The antimicrobial assay showed similar results, with high doses of NOS3 (MOI 100 to 1000) significantly reducing bacterial cell growth (Fig 4d). The baculovirus expressing NOS3 had a comparable response to 10 μM of nitrate. NOS3 and nitric oxide are also known for their role in cardiovascular health specifically regulating blood pressure through vasodilation, preventing endothelial dysfunction, and reducing arterial stiffness [43]. Even at a MOIs of 1000, no difference in cell proliferation was observed with the EGFP-baculovirus. Comparatively, high amounts of the NOS3-baculovirus significantly reduced SMC proliferation. An MOI of 100 and 1000 of NOS3 also increased HUVEC proliferation without impacting viability (Fig 4e and 4f).

### Cytotoxicity of the PVA-based hydrogels eluting NOS3 expressing baculovirus

The baculovirus was non-cytotoxic in SMCs and HUVECs, even in high amounts (Fig 5). Different amounts (MOIs) of the baculovirus were tested, ranging from one to 1000. Even at a MOIs of 1000, no difference in viability was observed with the baculovirus. The PVA-based hydrogels were also incubated directly with SMCs or HUVECs. Similarly, the PVA-based hydrogels were non-cytotoxic to SMCs and HUVECs. The hydrogels did not significantly reduce cell proliferation or viability compared to the mock cells (Fig 5). The PVA-PEG hydrogel reduced the cell viability the most after 48 hours. Comparatively, the PVA and PVA-gelatin hydrogels maintained the highest cell viability and proliferation.

The PVA baculovirus delivery system exhibits excellent safety properties. Specifically, no significant difference in cell proliferation or viability after transduction with baculovirus in both SMCs and HUVECs was observed. Even at MOIs above 1000, no cell proliferation or viability change was noted with the EGFP-baculovirus. A significant decrease in SMC proliferation was observed with higher NOS3 levels as expected [12]. Similarly, no significant difference in viability or proliferation was observed after incubation with the different hydrogels. This indicates that the baculovirus-eluting PVA delivery system does not pose cytotoxic effects on the cells. This study supports previous research reporting that baculoviruses and PVA hydrogels are non-toxic to mammalian cells [2, 44]. Future work can focus on specific clinical applications to optimize the baculovirus delivery system. Moreover, the customized system could be tested *in vivo* to confirm its therapeutic application.

Depending on the required therapeutic time frame and cell type, the baculovirus elution amount and time can be customized for each application. The PVA-based hydrogels presented here would be suitable for medical device coatings. Specifically, PVA-gelatin stent coatings eluting NOS3 would be beneficial due to the antithrombotic properties, safety, and sustained gene delivery. This controlled baculovirus delivery system can be tailored to the countless diseases requiring gene or growth factor delivery. Similarly, adding another polymer, such as chitosan, gelatin, or PEG, can alter the PVA properties. To our knowledge, this is the first study optimizing PVA-based hydrogels containing baculovirus gene delivery vectors.

### Conclusion

PVA-based hydrogel delivery of baculoviruses to mammalian cells is efficient, safe, and customizable. The hydrogel delivery system exhibited good blood compatibility and was non-cytotoxic to SMCs and HUVECs. The baculovirus loading efficiency reached 96.4 ± 0.6% and

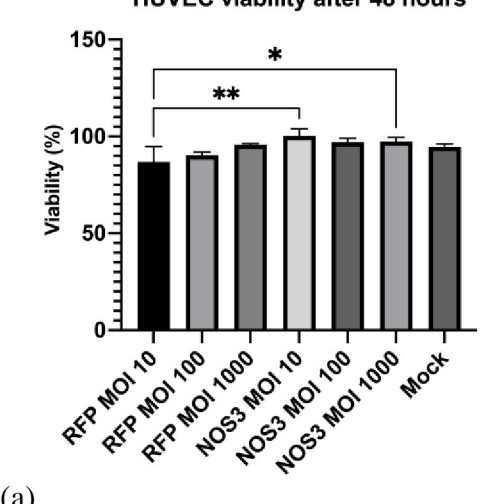

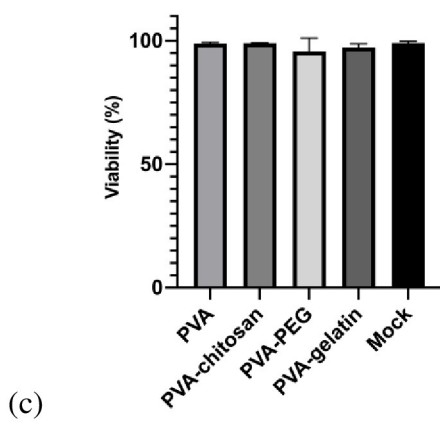

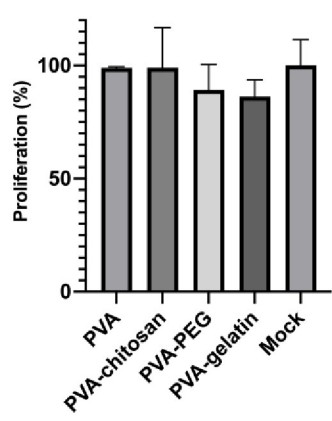

**Fig 5. Mammalian cell viability after baculovirus transduction with different MOIs or direct hydrogel incubation.** (a) HUVEC viability after BV transduction. (b) SMC viability after BV transduction. (c) HUVEC viability after incubating with different hydrogels for 48 hours. (d) HUVEC proliferation after hydrogel incubation for 48 hours.

was sustained for up to 25 days. The delivery system can be further customized by adding other polymers, loading methods, or additional freeze-thaw cycles. These alterations can change the hydrogel's mechanical properties and gene elution, specific for each therapeutic application.

## Supporting information

**S1 Data.**
(XLSX)

## Acknowledgments

We thank the Facility for Electron Microscopy Research of McGill University (SEM equipment), McGill Chemistry Characterization Facility (ATR-FTIR), and the McGill University

Imaging and Molecular Biology Platform (IMBP, Operetta High Content microscope) for equipment usage and services.

## Author Contributions

**Conceptualization:** Sabrina Schaly.

**Data curation:** Sabrina Schaly, Paromita Islam, Rahul Thareja.

**Formal analysis:** Sabrina Schaly, Rahul Thareja, Karan Arora.

**Investigation:** Sabrina Schaly.

**Methodology:** Sabrina Schaly, Paromita Islam, Jacqueline L. Boyajian, Ahmed Abosalha, Karan Arora, Satya Prakash.

**Project administration:** Satya Prakash.

**Resources:** Dominique Shum-Tim, Satya Prakash.

**Software:** Ahmed Abosalha.

**Supervision:** Dominique Shum-Tim.

**Writing – original draft:** Sabrina Schaly.

**Writing – review & editing:** Jacqueline L. Boyajian, Satya Prakash.

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
