## [Decision Letter · Decision Letter 0]

22 May 2023

PONE-D-23-10026Controlled and customizable baculovirus NOS3 gene delivery using PVA-based hydrogel systems.PLOS ONE

Dear Dr. Prakash,

Thank you for submitting your manuscript to PLOS ONE. After careful consideration, we feel that it has merit but does not fully meet PLOS ONE’s publication criteria as it currently stands. Therefore, we invite you to submit a revised version of the manuscript that addresses the points raised during the review process.

We look forward to receiving your revised manuscript.

Kind regards,

Jian Xu, Ph.D.

Academic Editor

PLOS ONE

Journal Requirements:

"YES. This work was supported by a grant from the Canadian Institute of Health Research (CIHR) to Dominique Shum-Tim and Satya Prakash. S.S. is fully funded by the Canadian Graduate Scholarship-Doctoral from the Natural Sciences and Engineering Research Council (NSERC). P.I. is funded by the Islamic Development Bank Scholarship (2020-245622). A.A. is fully funded by a scholarship from the Ministry of Higher Education of the Arab Republic of Egypt. R.T. and K.A. are funded by the Canadian Graduate Scholarship-Master’s from NSERC."

"NO authors have competing interests"

Reviewers' comments:

Reviewer's Responses to Questions

**Comments to the Author**

1. Is the manuscript technically sound, and do the data support the conclusions?

Reviewer #1: Partly

Reviewer #2: Yes

2. Has the statistical analysis been performed appropriately and rigorously? 

Reviewer #1: No

Reviewer #2: Yes

3. Have the authors made all data underlying the findings in their manuscript fully available?

Reviewer #1: Yes

Reviewer #2: Yes

4. Is the manuscript presented in an intelligible fashion and written in standard English?

Reviewer #1: No

Reviewer #2: Yes

5. Review Comments to the Author

Reviewer #1: The study provides important insights into the potential of NOS3 eluting PVA-based hydrogels for medical applications and device coatings. The addition of gelatin to the hydrogels is shown to be effective in protecting baculovirus viability and enabling sustained release. However, the study has limitations in terms of investigating the actual effects of NOS3 delivery on cell proliferation, blood vessel vasodilation, and bacterial growth inhibition, as well as providing detailed characterization of the properties of the hydrogels. Nevertheless, the study has value in providing a proof of concept for a potential delivery system and highlights areas for further research and development.

The study only investigates the release of baculoviruses and their potential for NOS3 gene delivery. It does not investigate the actual effect of NOS3 delivery on cell proliferation, blood vessel vasodilation, or bacterial growth inhibition.

The study only investigates the effect of hydrogel composition on properties such as elasticity, blood coagulation, and antimicrobial effects, but does not provide a detailed characterization of these properties.

The study does not investigate the long-term stability of the hydrogels or the effect of storage conditions on their properties and performance.

Reviewer #2: This manuscript evaluates the use of hydrogel formulations for sustained delivery of baculovirus in a proof-of-concept study. The manuscript is well written, properly illustrated and appropriately referenced. The result stand on their own. Baculovirus as a vehicle for gene delivery and gene editing is increasingly gaining interest and traction due to the numerous advantages of this viral vector above the state-of-the-art. This study represents and interesting and insightful contribution that will be valuable for the field.

6. PLOS authors have the option to publish the peer review history of their article (what does this mean?). If published, this will include your full peer review and any attached files.

Reviewer #1: No

Reviewer #2: No

---

## [Author Response · Author response to Decision Letter 0]

25 Jul 2023

PONE-D-23-10026 Editor & Reviewer Responses 

Controlled and customizable baculovirus NOS3 gene delivery using PVA-based hydrogel systems

Dear Editor:

Thank you for your advice and recommendations. We appreciate the opportunity to revise manuscript. We have followed and completed all your recommendations. Briefly: 

1. We have reformatted the manuscript and figure files to meet PLOS One’s requirements. 

2. The funding information has been checked and should be as follows:

a. This work was supported by a grant from the Canadian Institute of Health Research (CIHR, grant 252743) to Dominique Shum-Tim and Satya Prakash. S.S. is fully funded by the Canadian Graduate Scholarship-Doctoral from the Natural Sciences and Engineering Research Council (NSERC, 569661-2022).). P.I. is funded by the Islamic Development Bank Scholarship (2020-245622). A.A. is fully funded by a scholarship from the Ministry of Higher Education of the Arab Republic of Egypt. R.T. and K.A. are funded by the Canadian Graduate Scholarship-Master’s from NSERC. The funders had no role in study design, data collection and analysis, decision to publish, or preparation of the manuscript.

3. The statement has been added after the funding information provided in the manuscript. 

4. The competing interests form has been completed, http://journals.plos.org/plosone/s/submit-now

5. Data availability statement: the minimal data underlying the results has been provided in the attached Excel sheet with each figure labelled. 

6. Ethics approval was not required for the study and this statement has been added to the Methods section of the manuscript. 

Reviewer #1

Comment: The study provides important insights into the potential of NOS3 eluting PVA-based hydrogels for medical applications and device coatings. The addition of gelatin to the hydrogels is shown to be effective in protecting baculovirus viability and enabling sustained release. However, the study has limitations in terms of investigating the actual effects of NOS3 delivery on cell proliferation, blood vessel vasodilation, and bacterial growth inhibition, as well as providing detailed characterization of the properties of the hydrogels. Nevertheless, the study has value in providing a proof of concept for a potential delivery system and highlights areas for further research and development.

The study only investigates the release of baculoviruses and their potential for NOS3 gene delivery. It does not investigate the actual effect of NOS3 delivery on cell proliferation, blood vessel vasodilation, or bacterial growth inhibition.

The study only investigates the effect of hydrogel composition on properties such as elasticity, blood coagulation, and antimicrobial effects, but does not provide a detailed characterization of these properties.

The study does not investigate the long-term stability of the hydrogels or the effect of storage conditions on their properties and performance.

Response: Thank you for your approval of the manuscript. We also appreciate your important suggestions. As per your suggestions and recommendations, we have revised this manuscript. You will find we have addressed all your concerns. Briefly:

The effect of NOS3 delivery on both HUVEC and SMC proliferation is presented in Figure 4e-f. We have also added a Griess assay to measure nitrate concentration (Figure 4d). An antibacterial assay using NOS3 and nitrate has been added. Moreover, previous papers have already established NOS3 and nitric oxide as a cardiovascular support and antimicrobial properties and have been cited in lines 470-478. 

Further details have also been added for the hydrogel swelling, antimicrobial effects, and blood coagulation on pages 13-14. 

The long-term stability was added to lines 456-460 and 334-335. 

Storage conditions of the baculovirus-eluting hydrogels have been added to lines 456-460.

Reviewer #2:

This manuscript evaluates the use of hydrogel formulations for sustained delivery of baculovirus in a proof-of-concept study. The manuscript is well written, properly illustrated and appropriately referenced. The result stand on their own. Baculovirus as a vehicle for gene delivery and gene editing is increasingly gaining interest and traction due to the numerous advantages of this viral vector above the state-of-the-art. This study represents and interesting and insightful contribution that will be valuable for the field.

Response: Thank you for your acceptance and recommendation of the article for publication.

---

## [Decision Letter · Decision Letter 1]

18 Aug 2023

Controlled and customizable baculovirus NOS3 gene delivery using PVA-based hydrogel systems.

PONE-D-23-10026R1

Dear Dr. Prakash,

We’re pleased to inform you that your manuscript has been judged scientifically suitable for publication and will be formally accepted for publication once it meets all outstanding technical requirements.

Kind regards,

Jian Xu, Ph.D.

Academic Editor

PLOS ONE

Additional Editor Comments (optional):

Reviewers' comments:

Reviewer's Responses to Questions

**Comments to the Author**

1. If the authors have adequately addressed your comments raised in a previous round of review and you feel that this manuscript is now acceptable for publication, you may indicate that here to bypass the “Comments to the Author” section, enter your conflict of interest statement in the “Confidential to Editor” section, and submit your "Accept" recommendation.

Reviewer #1: All comments have been addressed

2. Is the manuscript technically sound, and do the data support the conclusions?

Reviewer #1: Yes

3. Has the statistical analysis been performed appropriately and rigorously? 

Reviewer #1: Yes

4. Have the authors made all data underlying the findings in their manuscript fully available?

Reviewer #1: Yes

5. Is the manuscript presented in an intelligible fashion and written in standard English?

Reviewer #1: Yes

6. Review Comments to the Author

Reviewer #1: The authors have addressed the comments. The manuscript can be considered for publication in its present form.

7. PLOS authors have the option to publish the peer review history of their article (what does this mean?). If published, this will include your full peer review and any attached files.

Reviewer #1: No

---

## [Editor Report · Acceptance letter]

13 Sep 2023

PONE-D-23-10026R1 

Controlled and customizable baculovirus NOS3 gene delivery using PVA-based hydrogel systems 

Dear Dr. Prakash:

I'm pleased to inform you that your manuscript has been deemed suitable for publication in PLOS ONE. Congratulations! Your manuscript is now with our production department. 

Kind regards, 

on behalf of

Dr. Jian Xu 

Academic Editor

PLOS ONE